# Path Generation Strategy and Wire Arc Additive Manufacturing of Large Aviation Die with Complex Gradient Structure

**DOI:** 10.3390/ma15176115

**Published:** 2022-09-02

**Authors:** Jiansheng Zhang, Guiqian Xiao, Jie Peng, Yingyan Yu, Jie Zhou

**Affiliations:** 1Chongqing Key Laboratory of Advanced Mold Intelligent Manufacturing, College of Materials Science and Engineering, Chongqing University, Chongqing 400044, China; 2Chongqing Jiepin Technology Co., Ltd., Chongqing 400000, China

**Keywords:** WAAM, contour-parallel path generation strategy, large aviation die, application test

## Abstract

To realize automatic wire arc additive manufacturing (WAAM) of a large aviation die with a complex gradient structure, a new contour-parallel path generation strategy was proposed and practically applied. First, the planar curve was defined as a vertical slice of a higher-dimensional surface and a partial differential equation describing boundary evolution was derived to calculate the surface. The improved Finite Element Method (FEM) and Finite Difference Method (FDM) were used to solve this partial differential equation. Second, a cross section of a large aviation die was used to test the path-generation algorithms. The results show that FEM has a faster solving speed than FDM under the same solving accuracy because the solving domain of FEM mesh was greatly reduced and the boundary mesh could be refined. Third, the die was divided into three layers: base layer, transition layer (Fe-based material) and strengthening layer (Co-based material) according to the difference of the temperature and stress field, and corresponding WAAM process parameters has been discussed. The optimum welding parameters are obtained as follows: voltage is 28 V, wire feeding speed is 8000 mm/min and welding speed is 450 mm/min. Finally, the path generation strategy was practically applied to the remanufacture of the large aircraft landing gear die with a three-layer structure. The application test on aircraft landing gear dies justified the effectiveness of the algorithms and strategy proposed in this paper, which significantly improved the efficiency of the WAAM process and the service life of large aviation dies with complex gradient structures. The microstructure of the fusion zone shows that the base metal and welding material can be fully integrated into the welding process.

## 1. Introduction

The service life of traditional large aviation dies is extremely low due to severe working conditions with high temperatures and heavy loads. WAAM technology is gradually applied to the manufacturing and repairing of large aviation dies. The core tasks of WAAM are the welding path generation and application experiment, which directly affect the quality of additive manufacturing [1,2]. There is much research on welding path generation. Direction-parallel path and contour-parallel path are two basic methods for filling arbitrary plane boundaries. The contour-parallel path is more widely used for some complex boundaries’ WAAM because of its smoother filling trajectory and fewer turning points [3,4,5]. In addition, there are many other methods that have certain limitations and cannot be used for arbitrary boundaries to generate the filling path, such as raster [6], zigzag [7], spiral [8], fractal space-filling curves [9,10], and hybrid tool path planning approaches [11,12]. In the contour-parallel path, the contour offsetting will encounter many polygon operation problems [13], such as ambiguous normal (Figure 1a), singular point (Figure 1b), self-intersection (Figure 1c), change in topology (Figure 1d) and offsets intersection of pocket and island boundary (Figure 1e). Many approximate methods, such as iterative methods for spline offsetting [14,15], Voronoi diagram-based offsetting [16,17], pixel simulation and image processing-based offsetting [18] and Laplace-based formulation offsetting [19] have been proposed. In iterative methods for spline offsetting, the detection of singularities and redundancies is very complicated to carry out. Although complex detection can be avoided in Voronoi diagram-based offsetting, two-dimensional Boolean set operations take a long time to compute. Pixel simulation and image-processing-based offsetting require complex pre-processing and post-processing. Laplace-based methods are suitable for smooth and continuous path generation [20]. However, the part is sometimes overmachined, which depends on the shape and the distance between two neighboring contours [11].

In addition to filling algorithms, many scholars have conducted some application research on the WAAM process. Lu proposed a method for repairing forging dies with manual WAAM [21]. The surface strength and hardness of the die are strengthened by covering the high-strength material on the surface of the die. An automatic WAAM technology is proposed by Hong et al. [22] to realize the gradient additive remanufacturing of ultra-large hot forging dies. The experimental results show that the service life of the hot forging dies repaired by the automatic WAAM technology is significantly increased, the material is reduced by more than 50% and the production efficiency is increased by more than 50%. To improve the service life of hot forging dies, additive manufacturing algorithms and additive manufacturing devices for die remanufacturing have been developed [23]. The results show that using the techniques, welding materials can be saved by more than 50% and machining costs can be saved by more than 60%. The experimental results show that WAAM technology is very promising for mold repair, and its core lies in path planning.

To overcome the shortcomings of the above path generation methods, a new contour-parallel path generation strategy based on the Level Set Function (LSF) is proposed. First, a Partial Differential Equation (PDE) is constructed to describe the movement of the boundaries. The equidistant filling of arbitrary boundaries can be transformed into the equidistant movement of the initial boundary in which the moving fronts are expanded or contracted. Second, an improved Finite Element Method (FEM) and an improved Finite Difference Method (FDM) are developed to solve the partial differential equation of boundary value. Compared with FDM, FEM can obtain better solution accuracy with fewer nodes because the mesh of FEM can approximate the complex boundary with a smaller number of elements. Finally, a large aviation die is used to verify the new contour-parallel path generation strategy. The WAAM of the landing gear die proves that the algorithm and gradient structure proposed in this paper can improve the service life of the large aviation die and can improve the efficiency of the WAAM process simultaneously.

## 2. Contour-Parallel Path Generation Strategy

### 2.1. Mathematical Foundation

In general, a closed curve in a plane can be defined as $S\left(k\right)={\left[x\left(k\right),y\left(k\right)\right]}^{T}$, in which *k* is a parametrization variable. The offset curve of the closed curve is defined as follows:(1)$$S^{\prime }\left(k\right)=S\left(k\right)+\vec{n}d$$
where, $\vec{n}$ is the unit normal vector to the original curve at parametric point *k*. As discussed in the introduction, many problems arise with this approach, such as ambiguous normal, rarefaction fan at corner point, self-intersection, generation of singular point, change in topology and intersection of offsets of pocket and island boundary (Figure 1).

Another important way to define a closed curve in a plane is the intersection of a surface and a plane, as follows:(2)ψ(x,y)=0
where, ψ is an implicit function, also known as the level set function.

As shown in Figure 2a, the workpiece is cut to obtain a cross section where the inner boundary and outer boundary are defined as clockwise and anticlockwise arrangements, respectively. In particular, as shown in Figure 2b, the inner and outer boundaries can be represented by the 0-value contour of a three-dimensional surface *ψ*. The key task is how to construct a calculation method for this surface.

It is impossible to obtain the level set function directly. A desirable method is to determine the differential equation of the surface function. The time differential expression of Equation (2) is as follows:(3)$$\frac{\partial \psi }{\partial t}+F||\nabla \psi ||=0$$
where *F* = 1 − εγ and the γ is the curvature of the corresponding point on the curve. The ε is the smoothing term coefficient (for equidistant filling, ε = 0). *F* is the front speed in the direction normal to the boundary. Then, Equation (3) is a standard Hamilton-Jacobi equation [24]. The initial conditions on *ψ* are defined as follows:(4)$$\psi \left(x,y,t=0\right)=\pm min\;\sqrt{{\left(x-x^{\prime }\right)}^{2}+{\left(y-y^{\prime }\right)}^{2}}$$
where *x* and *y* represent the horizontal and vertical ordinates of any point in the plane, respectively. $x^{\prime }$ and $y^{\prime }$ represent the horizontal and vertical ordinates of any point on the boundary, respectively. It is worth noting that when the point is on the boundary, the function value *ψ* is equal to zero. As shown in Figure 3, the value of ψ(x,y,t=0) is more than zero when the point is in the cross section and less than zero when the point is outside the cross section.

If point $P^{\prime \prime }\left(x^{\prime \prime },y^{\prime \prime }\right)$ is the closest point to point P(x,y), the closest point to any point on the line $P^{\prime }P$ is $P^{\prime }$. Then, the following formula can be derived:(5)$$\nabla \psi \left(x,y,t=0\right)=\left(\frac{\partial \left(\sqrt{{\left(x-x^{\prime }\right)}^{2}+{\left(y-y^{\prime }\right)}^{2}}\right)}{\partial x},\frac{\partial \left(\sqrt{{\left(x-x^{\prime }\right)}^{2}+{\left(y-y^{\prime }\right)}^{2}}\right)}{\partial y}\right)$$

If the initial value of *ψ* is defined as Equation (4), it is easy to get ||∇ψ(x,y,t=0)||=1 from Equation (5). Therefore, Equation (3) can be simplified as follows:(6)$$\frac{\partial \psi }{\partial t}+F=0$$
where *F* is the distance from which the boundary moves in unit time.

### 2.2. Solution Algorithms

The mathematical foundation of the contour-parallel path generation strategy was introduced in the previous section. The boundary is usually non-analytical in actual contour-parallel path generation. Therefore, the non-analytical algorithm of Equation (6) needs to be derived. Generally, there are two methods for the numerical solution of partial differential Equation (6): the Finite Difference Method (FDM) and the Finite Element Method (FEM).

As shown in Figure 4, the solution domain is different for FDM and FEM. The solution domain of the FDM is the maximum envelope rectangle with boundaries in which the solution domain is discretized into a square lattice of n × n. The solution domain of FEM can be an irregular boundary, in which the solution domain is discretized into multiple triangles. When solving the same filling problem, the solution domain of FDM is larger than that of FEM. From this point of view, FEM is more efficient than FDM. The space complexity of FDM is O(n2). The space complexity of FEM is O(n). Therefore, from the perspective of solving space and mesh refinement, the efficiency of FEM will be higher than that of FDM.

The solution domain of FDM may be significantly larger than that of FEM for some special filled boundaries. In the case shown in Figure 4, the solution domain of FDM is nearly twice as large as that of FEM. Next, we explore the iterative formula in the solution domain to solve Equation (6). The iterative formulas of FEM and FDM differ. In our previous research [25], the iterative formula of FDM has been derived as follows:(7)$$\frac{\psi \left(x_{i},y_{j},t_{k+1}\right)-\psi \left(x_{i},y_{j},t_{k}\right)}{\Delta t}+F=0$$
where, $t_{k+1}=t_{k}+\Delta t$, $x_{i+1}=x_{i}+\Delta x$ and $y_{i+1}=y_{i}+\Delta y$. ∆*t*, ∆*x* and ∆*y* are increments in time, *x* and *y*, respectively. The smaller these values are, the higher the solution precision. In addition, *i* = 1, 2, …., M. *j* = 1, 2, …, N. *k* = 1, 2, …, w. The initial value of the level set function *ψ* can be calculated using Formula (4). $x_{i}$, $y_{j}$ and $t_{k}$ are abscissa, ordinate, and time of grid point (*i, j*), respectively. Similarly, the iterative formula of FEM can be derived as follows:(8)$$\frac{\psi \left(x_{i},y_{i},t_{k+1}\right)-\psi \left(x_{i},y_{i},t_{k}\right)}{\Delta t}+F=0$$
where, $x_{i}$, $y_{i}$ are abscissa and ordinate of node *i* (*i* = 1, 2, …., N), respectively. N represents the total number of nodes in the solution domain. As we can see from Formula (8), there is no coupling relationship between nodes. Then, the iteration formula is carried out only in the time dimension of each node. To simplify and improve the calculation speed, the linear shape function is used in FEM.

### 2.3. Computation Procedures

The solution flow of the FDM is shown in Figure 5a. The solution process can be divided into the following steps:

Step 1: Input the ∆*x*, ∆*y*, ∆*t*, *F*, Fill spacing and boundaries of the cross-section.

Step 2: Generate the FDM grid according to the boundaries.

Step 3: Calculate the initial value of the level set function *ψ* by using Equation (4).

Step 4: Calculate the level set function value *ψ* at the next time by using Equation (7).

Step 5: Judge whether the actual filling spacing is equal to the required filling spacing. If not, jump to step 4. If yes, continue to the next step.

Step 6: Calculate $\psi \left(x_{i},y_{j},t_{i}\right)=0$ by the contour calculation method.

Step 7: Reassemble the filling lines ($\psi \left(x_{i},y_{j},t_{i}\right)=0$) and store the contour-parallel path.

Step 8: Judge whether the equidistant filling is completed. If yes, continue to the next step. If not, jump to step 2 and replace the boundaries with the contour parallel path.

Step 9: Output the equidistant filling contour (contour-parallel path) and end the program.

Similarly, the solution flow of FEM is shown in Figure 6b. The solution process can be divided into the following steps:

Step 1: Input the time increment ∆*t*, the moving speed F, the element size, and the boundaries of the cross section.

Step 2: Generate FEM mesh with or without boundary refinement.

Step 3: Calculate the initial value of the level set function *ψ* for each node by using Equation (4).

Step 4: Calculate the level set function value *ψ* at the next time by using Equation (8).

Step 5: Judge whether the actual filling spacing is equal to the required filling spacing. If not, jump to step 4. If yes, continue to the next step.

Step 6: Calculate $\psi \left(x_{i},y_{j},t_{i}\right)=0$ by using the slicing method [26].

Step 7: Reassemble the filling lines ($\psi \left(x_{i},y_{j},t_{i}\right)=0$) and stores the contour-parallel path.

Step 8: Judge whether the equidistant filling is completed. If yes, continue to the next step. If not, jump to step 2 and replace the boundaries with the contour parallel path.

Step 9: Output the equidistant filling contour (contour-parallel path) and end the program.

The contour represented by $\psi \left(x_{i},y_{j},t_{i}\right)=0$ can be calculated with a mature STL model slicing algorithm. Using the value of $\psi \left(x_{i},y_{j},t_{i}\right)$ as the coordinate component in the height direction of the triangular mesh node in a plane and a new STL model, which can be sliced to get the zero-value contour, is obtained. Triangles close to the boundary can be refined to improve calculation accuracy with an equal number of triangles in the FEM. In FDM, the size of the quadrilateral mesh cannot be refined locally.

As shown in Figure 6, the contour-parallel path of a landing gear die has been generated by using FDM. The x-axes and y-axes describe the sections to be filled, the color represents the value of the level set function, and the solid red line indicates the filling line after the offset. After generating the paths, they are used as a new cross-section boundary to generate a new FDM grid. With the increase in offsetting times, the grid area remains the same. Therefore, the computation time is not changed.

As shown in Figure 7, the contour-parallel path of a landing gear die has been generated by using FEM. After generating the paths, they are used as a new cross-section boundary to generate a new FEM mesh. With the increase in offsetting times, the mesh area becomes smaller and smaller until it disappears. Therefore, the computation time is shorter and shorter.

The comparison of the path generation processes of FEM and FDM (Figure 6 and Figure 7) shows that FEM can significantly reduce the calculation time. There are two reasons why the FEM algorithm is faster than the FDM algorithm. First, the solution domain of FEM can be greatly reduced with an increase in the number of offsets (only in the filled area), while the solution domain of FDM does not decrease with an increase in the number of offsets (in the largest rectangular area of the entire cross-section). Second, the mesh of the FEM can be refined at the boundary, which can ensure that fewer elements are used with the same accuracy compared with FDM, in which the element cannot be refined at the boundary.

In addition, polygon splitting, disappearing and merging with boundary migration can evolve automatically without special processing by using these two solving algorithms. As the example shown in Figure 6 and Figure 7, connected domains evolve from three to one, then to five, then to six and finally disappear. It is very complicated to deal with this evolution with conventional geometric offsetting algorithms. As we can see from Figure 8, there are subtle differences between the equidistant offset filling contours generated by FDM and FEM. The contours obtained by these two algorithms can meet the requirements of the WAAM process. FEM can reduce the solving domain and thus has higher efficiency. However, the FDM grid is simple and easy to generate. The FEM mesh is slightly more complex and mature mesh generation algorithms can be used. Although many types of paths are used in WAAM, such as raster, zigzag, spiral, fractal space-filling curves, and hybrid toolpath planning approaches, the contour-parallel path is the most appropriate for this process. Simultaneously, the continuous smooth contour-parallel path can reduce drastic changes in the velocity and acceleration of the robot.

## 3. Application Test

### 3.1. Gradient Structure Design

This section introduces a practical application example of the proposed algorithms in the repair process of a failed landing gear die using WAAM technology. The repair process includes modeling, slicing, single-layer contour-parallel path generation, robot code generation, simulating and WAAM. The failed die is repaired accurately with different materials for different areas, thus realizing the equal-life design concept.

As shown in Figure 9, different areas of the landing gear die have different temperatures and stress distributions during service. Therefore, it is necessary to design different materials for different areas before the repair process, which is also called gradient structure design (GSD). Due to the high temperature and high stress of the surface layer, Co-based materials are chosen. The transition layer can be deposited with Fe-based materials. The material of the die base is 5GrNiMo.

### 3.2. WAAM Process Parameters

The quality of a single weld pass is one of the core factors affecting the entire quality of the welding plane. A simple welding test is shown in Figure 10, in which the welding voltage is 28 V and the welding wire diameter is 1.6 mm. As shown in Figure 11, with the wire feeding speed (WFS) getting larger or the walking speed (WS) getting smaller, the weld pass becomes wider. If higher welding accuracy is required, the width of the weld pass should be smaller, which will reduce the welding efficiency. In this paper, the size of the aviation die to be repaired is large and to improve the WAAM efficiency, a larger WFS is selected.

As shown in Figure 12a–c, when the wire feeding speed and the welding voltage are constant, with the increase of welding speed, the linear energy density decreases and the cooling rate of the molten pool increases, forming fine columnar crystals, and the dendrite spacing decreases, which leads to the reduction of the columnar crystal size and the dendrite spacing. Conversely, as shown in Figure 12b,d, when the welding speed and the welding voltage are constant, with the increase in wire feeding speed, coarse columnar crystals are formed and the dendrite spacing increases. Then, from the perspective of microstructure, when the wire feeding speed and voltage are fixed, the larger the welding speed in a certain range, the better the microstructure.

To guarantee WAAM efficiency and error control in the height direction, weld height and walking speed should be chosen moderately. In this paper, the wire feeding speed is set to 8000 mm/min and the walking speed is set to 450 mm/min (Figure 10h and Figure 11h). According to corresponding research [22,27], the weld spacing is set to 1/2 of the weld width.

### 3.3. Repair Process of the Landing Gear Die

The steps to get the target WAAM model are scanning to obtain the points cloud, encapsulating the points cloud into a triangular mesh, converting the mesh into a parametric surface, making a difference with the standard model to obtain the temporary model, and cutting the temporary model according to the simulation results to get the target WAAM model. Similar steps have been explained in other research [22]. Then, the target model is sliced and the WAAM path is generated by the algorithms proposed above. A six-axis gantry robot has been developed for repairing large hot forging dies, in which a three-axis has been used to weld and the other three-axis has been used to hammer. After the welding of one layer is completed, this layer is hammered to strengthen the welding material and eliminate residual stress.

The repair process of the landing gear die is shown in Figure 13 and Co-based welding wire is used for the strengthening layer. The WAAM parameters are as follows: voltage is 28 V, wire feeding speed is 8000 mm/min and welding speed is 450 mm/min. As shown in Figure 13b, the welding process can achieve near no-splash, which enhances the fusion effect between welding materials and the die base.

As shown in Figure 13, by using the contour-parallel welding path, higher WAAM accuracy, less machining allowance and materials consumption are realized. In addition, as shown in Figure 13c, the machined surface is smooth without welding defects such as pores, slag inclusions and cracks, which can meet the needs of die repair.

### 3.4. Service Life Analysis

The high-temperature performance of the die was significantly improved after using Co-based material on the strengthening layer. For traditional dies, the flash land has obvious collapse and wear defects after twice forging (Figure 14a). On the contrary, for the repaired die with a gradient structure, the die surface is bright and no collapse and wear defects were found after 20 times of forging (Figure 14b). The service life of the aircraft landing gear die has been significantly improved by using WAAM technology with a gradient structure.

As shown in Figure 15, the fusion effect between the transition layer and the welding strengthening layer is good, and the microstructure presents a certain transition structure. The grain size gradually increases from the die base to the fusion line. This indicates that the heat input of the welding influences the grain size of the previous layer. In addition, the weld dendrites (Figure 12) near the fusion line are transformed into equiaxed crystals (Figure 16) after heat treatment. This fusion line indicates that the mechanical properties between the transition layer and the strengthening layer are well combined. The element distribution near the fusion line is shown in Figure 16. Due to the inconsistent composition, the fusion effect between welding material and base metal needs to be considered in the welding process. Cobalt and iron permeate each other around the fusion line. This shows that the base metal and welding material can be fully integrated into the welding process.

## 4. Conclusions

(1)In this paper, the mathematical basis of a new equidistant contour-parallel path generation strategy has been developed, and two types of iterative formulas based on FEM and FDM are proposed to solve the partial differential equation. By comparison, FEM has a faster solving speed than FDM under the same solving accuracy because the solving domain of FEM mesh can be greatly reduced, and the boundary mesh can be refined. However, the FDM grid is easier to generate and program than the FEM mesh. The two algorithms have their own advantages and disadvantages, and the selection of which algorithm can be determined according to the shape of the cross-section.(2)The die was divided into three layers: base layer, transition layer (Fe-based material) and strengthening layer (Co-based material) according to the temperature and stress field distribution of the aircraft landing gear die in service. The corresponding WAAM process parameters have been discussed. The optimum welding parameters are obtained as follows: voltage is 28 V, wire feeding speed is 8000 mm/min and welding speed is 450 mm/min. Finally, the path generation strategy was practically applied to the remanufacture of the large aircraft landing gear die with a three-layer structure.(3)The application test of the aircraft landing gear die justifies the effectiveness of the algorithms and technology proposed in this paper. The metallographic photos show that a good combination of die base and welding materials can be obtained at the micro level. In addition, using this new contour-parallel path generation strategy in the WAAM process can accurately control machining allowances and improve additive efficiency. The service life of the aircraft landing gear die can be significantly improved by using WAAM technology with a gradient structure.

## Figures and Tables

**Figure 1 materials-15-06115-f001:**
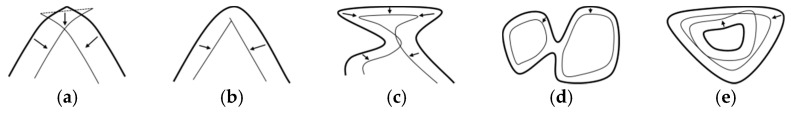
Various problems encountered in contour offsetting [11]: (**a**) Ambiguous normal; (**b**) Singular point; (**c**) Self-intersection; (**d**) Change in topology; (**e**) offset intersection of pocket and island boundary.

**Figure 2 materials-15-06115-f002:**
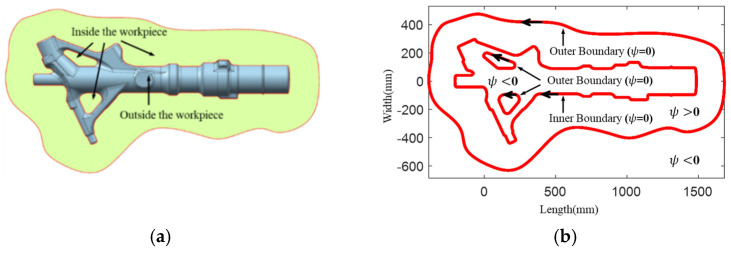
Cross section and boundary of a landing gear die: (**a**) Cross section; (**b**) Boundary.

**Figure 3 materials-15-06115-f003:**
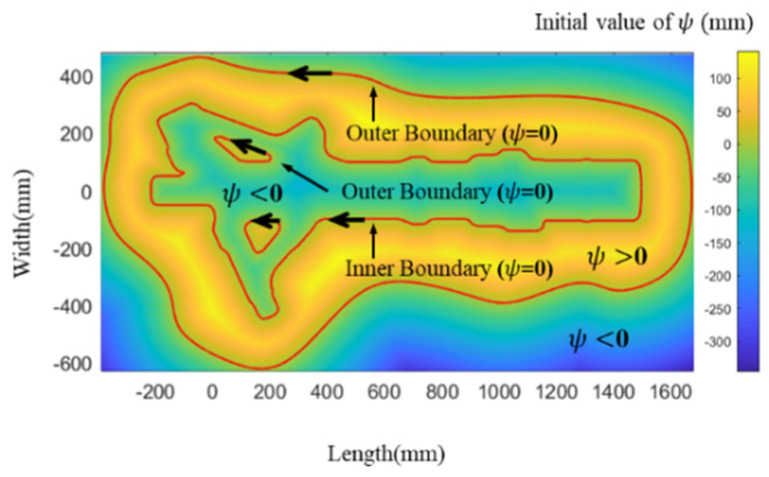
Initial value distribution of the level set function *ψ*.

**Figure 4 materials-15-06115-f004:**
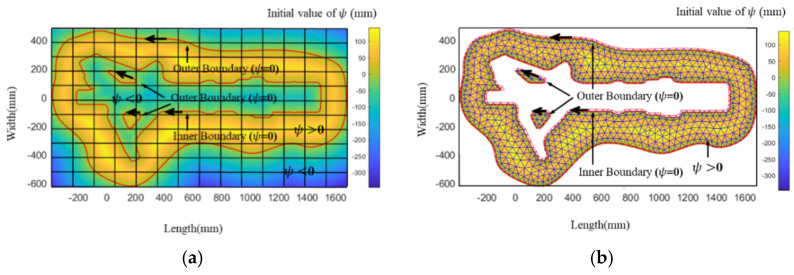
Solution domain and initial value of level set function *ψ*: (**a**) FDM; (**b**) FEM.

**Figure 5 materials-15-06115-f005:**
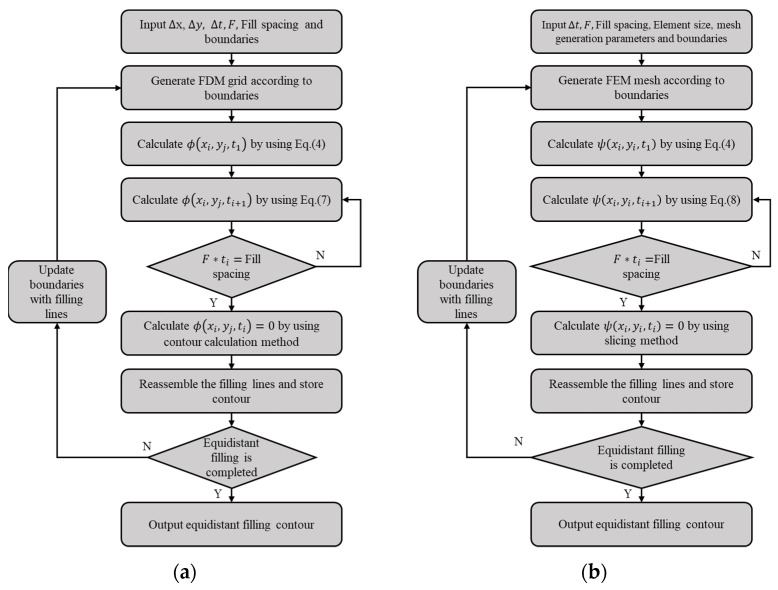
Flow chart of the new contour parallel path generation algorithm using different methods: (**a**) FDM; (**b**) FEM.

**Figure 6 materials-15-06115-f006:**
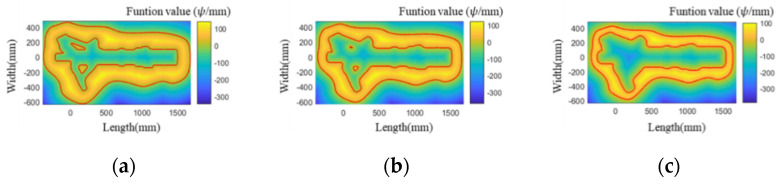
Contours and grids with different offset distances generated by using FDM: (**a**) 10 mm; (**b**) 30 mm; (**c**) 50 mm; (**d**) 70 mm; (**e**) 90 mm; (**f**) 110 mm.

**Figure 7 materials-15-06115-f007:**
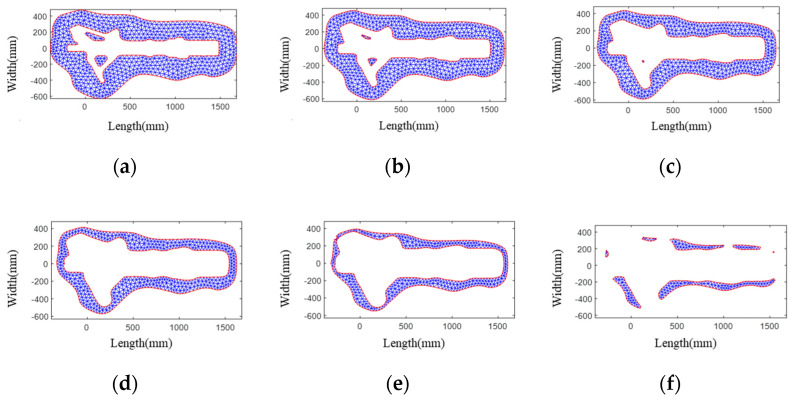
Contours and mesh with different offset distances generated by using FEM: (**a**) 10 mm; (**b**) 30 mm; (**c**) 50 mm; (**d**) 70 mm; (**e**) 90 mm; (**f**) 110 mm.

**Figure 8 materials-15-06115-f008:**
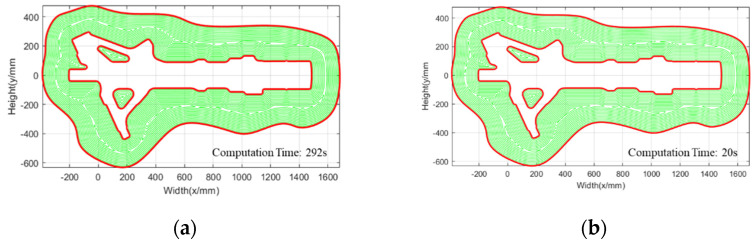
The equidistant offset filling contours by using different algorithms: (**a**) FDM; (**b**) FEM.

**Figure 9 materials-15-06115-f009:**
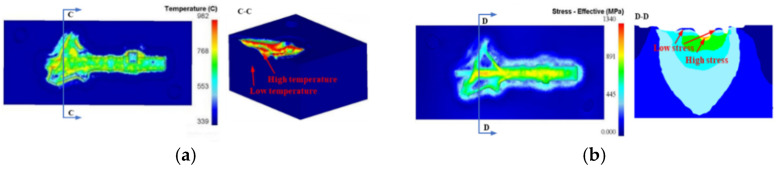
Stress and temperature field and gradient structure design: (**a**) Stress field; (**b**) Temperature field; (**c**) Gradient structure.

**Figure 10 materials-15-06115-f010:**
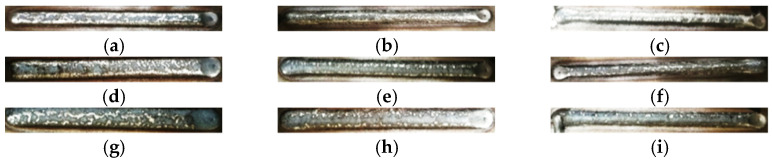
Weld passes obtained with different process parameters: (**a**) WS: 200 mm/min WFS: 4000 mm/min; (**b**) WS: 450 mm/min WFS: 4000 mm/min; (**c**) WS: 700 mm/min WFS: 4000 mm/min; (**d**) WS: 200 mm/min WFS: 6000 mm/min; (**e**) WS: 450 mm/min WFS: 6000 mm/min; (**f**) WS: 700 mm/min WFS: 6000 mm/min; (**g**) WS: 200 mm/min WFS: 8000 mm/min; (**h**) WS: 450 mm/min WFS: 8000 mm/min; (**i**) WS: 700 mm/min WFS: 8000 mm/min.

**Figure 11 materials-15-06115-f011:**
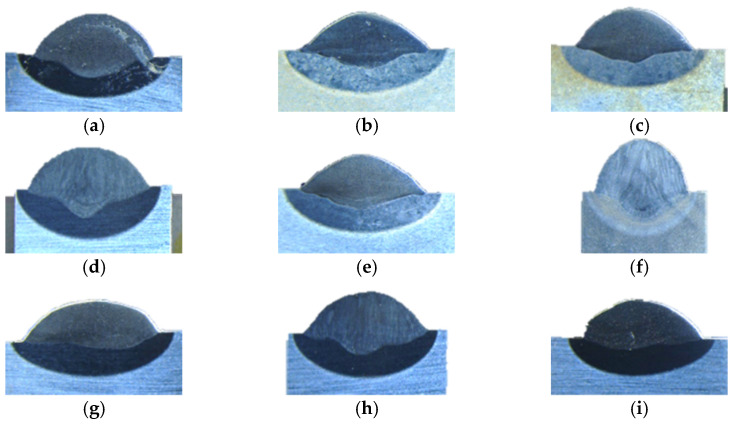
Morphology of weld passes obtained with different process parameters: (**a**) WS: 200 mm/min WFS: 4000 mm/min; (**b**) WS: 450 mm/min WFS: 4000 mm/min; (**c**) WS: 700 mm/min WFS: 4000 mm/min; (**d**) WS: 200 mm/min WFS: 6000 mm/min; (**e**) WS: 450 mm/min WFS: 6000 mm/min; (**f**) WS: 700 mm/min WFS: 6000 mm/min; (**g**) WS: 200 mm/min WFS: 8000 mm/min; (**h**) WS: 450 mm/min WFS: 8000 mm/min; (**i**) WS 700 mm/min WFS: 8000 mm/min.

**Figure 12 materials-15-06115-f012:**
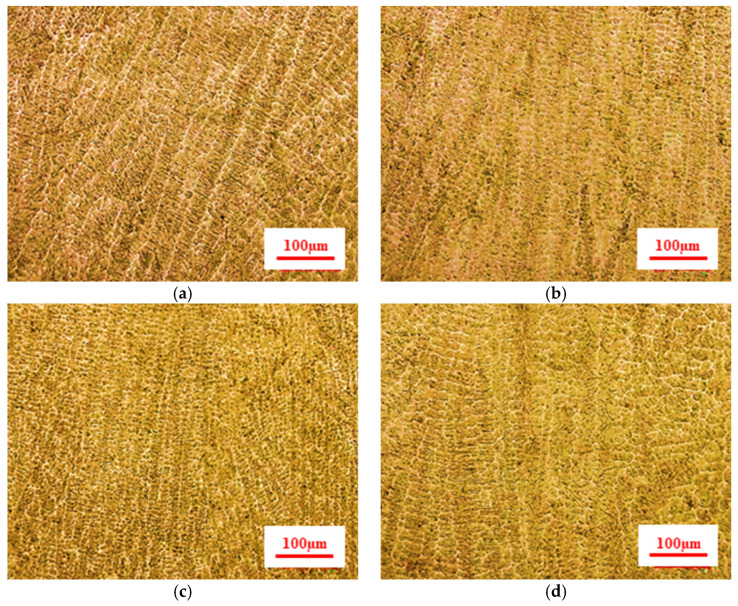
Microstructure of the weld pass with different linear energy densities: (**a**) WS: 200 mm/min WFS: 8000 mm/min; (**b**) WS: 450 mm/min WFS: 8000 mm/min; (**c**) WS: 700 mm/min WFS: 8000 mm/min; (**d**) WS: 450 mm/min WFS: 9000 mm/min.

**Figure 13 materials-15-06115-f013:**
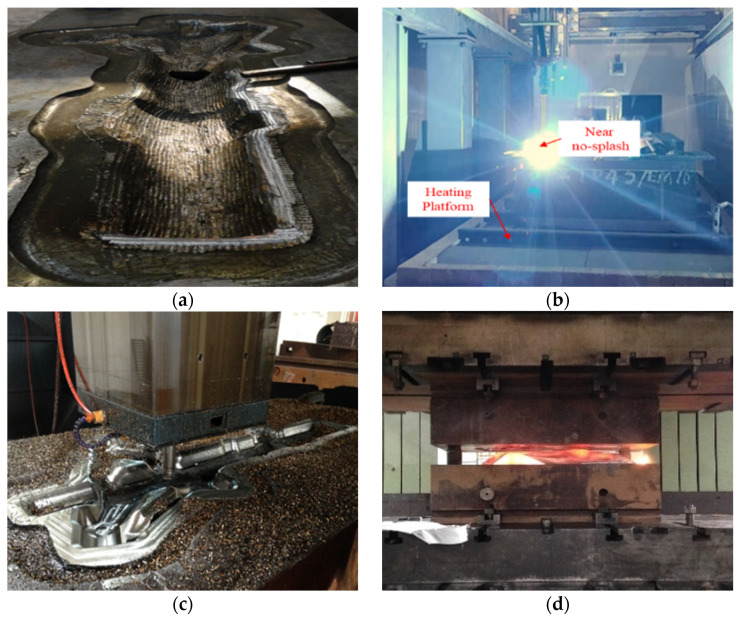
Repair process of the aircraft landing gear die using WAAM: (**a**) Remove the failure area; (**b**) During WAAM; (**c**) Die after CNC machining; (**d**) Service site.

**Figure 14 materials-15-06115-f014:**
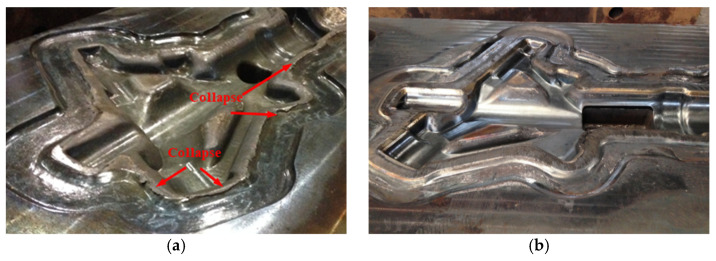
Appearance of traditional homogeneous die and repaired die with gradient structure: (**a**) Appearance of traditional homogeneous die after forging twice; (**b**) Appearance of gradient structure die after forging 20 times.

**Figure 15 materials-15-06115-f015:**
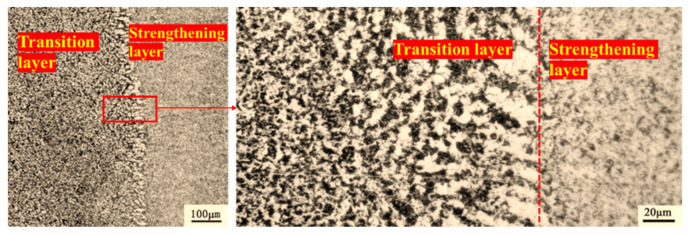
Microstructure of fusion zone between transition layer and strengthening layer.

**Figure 16 materials-15-06115-f016:**
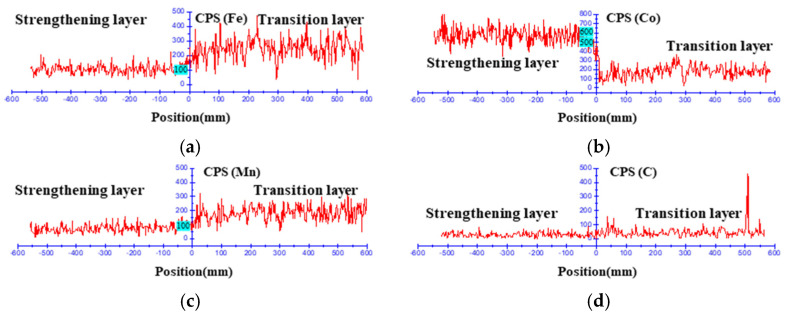
Element distribution near the fusion line: (**a**) Fe; (**b**) Co; (**c**) Mn; (**d**) C.

## Data Availability

Not applicable.

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
