# Peer review of "Path Generation Strategy and Wire Arc Additive Manufacturing of Large Aviation Die with Complex Gradient Structure"

_materials, 2022, doi:10.3390/ma15176115_

Round 1

Reviewer 1 Report

The manuscript titled "Path Generation Strategy and Wire Arc Additive Manufacturing of Large Aviation Die with Complex Gradient Structure" is a very interesting attempt. I appreciate the authors for sound planning and execution of the work.

I have few queries as listed below:

1. In abstract the experimental related analysis can be incorporated by reducing exclusive simulations.

2. Introduction can be improved further by discussing the experimental related work, which is already published related or close to this work.

3. The authors need to emphasize more on why FDM took a long time than FEM for the same scale, even the element size is reduced in FEM.

4. In Fig.6, (a) to (f) indicate variation in distance, whereas all the figures in Fig.6 indicate the x-axes as length. This need to be clearly explained so as to make it understandable to a common reader.

5. The authors need to explain the gradient structure design concept in their model. More details are needed in this aspect.

6. In Fig.12, the microstructural features need to be explained in detail.

7. Fig.13 the images are not really explainable what the authors are claiming.

8. More emphasis is needed on interface microstructures as provided in Fig.16.

9. The total number of figures is more. Maybe, the authors can try to merge the images wherever are applicable.

10. Conclusions can be improved further.

Reviewer 2 Report

This article deals with a contour-parallel path generation strategy, which is based on the construction of a partial differential equation, solved both by Finite Element Method and Finite Difference Method, with the aim of determining advantages and disvantages of the two algorithms.
This topic fits well into the current research scenario; however, the article suffers of some lacks.

First of all, it is necessary to update the bibliographical refererces in the introduction: e.g., regarding pixel generation the author may cite Ferreira and Scotti (10.3390/met11030498) and for a review Yaseer and Chen (10.1142/S0219686721500293).
Furthermore, a recent paper by Shen et al. (10.21203/rs.3.rs-285538/v1) has shown a path generation method for remanufacturing a damaged hot forging die, which can be useful for the authors to highlight the novelty of their work in the comparison.

The second aspect to be improved is of a methodological nature, in fact several features of the experimental setup are missed. The authors performed a scanning survey of the damaged die, probably by a laser system; so, more information is needed.  A brief description of the robotic apparatus and a map of the multi-bead welding profile are missed and a sketch of the process in more detail is desirable.  
The single pass beads, obtained under different welding conditions, are shown in fig. 11.  As for multipass effects, I advice to verify the cross-section profile in fig. 13 evaluating overlapping and empty areas ( for example, see the procedure carried out by Nguyen et al. (10.1016/j.promfg.2020.04.129) ).
Finally, I would like to read the compositions of the Co-based and Fe-based alloys that were used: which of the two materials do the micrographs in fig 12 refer to?

I think that, after having been revised, this article will deserve to be published.

Round 2

Reviewer 2 Report

In my opinion the reviewers' comments are well argued and the revised manuscript is suitable for publication.

However, figure 16 would be clearer if the strengthening layer side was labeled (I suppose it is on the left where the Co signal is highest, while in the micrographs it is on the right).

Finally, as a metallurgist I remain curious to know the composition of the used Fe and Co-based alloys (but it is probably confidential).
